# ReLoRA: High-Rank Training Through Low-Rank Updates

**Vladislav Lialin**[†,‡*] **Sherin Muckatira**[†], **Namrata Shivagunde**[†], **and Anna Rumshisky**[†,§]

[†]University of Massachusetts Lowell

[‡]Eleuther AI

[§]Amazon

## Abstract

Despite the dominance and effectiveness of scaling, resulting in large networks with hundreds of billions of parameters, the necessity to train overparameterized models remains poorly understood, while training costs grow exponentially. In this paper, we explore parameter-efficient training techniques as an approach to training large neural networks. We introduce a novel method called ReLoRA, which utilizes low-rank updates to train high-rank networks. We apply ReLoRA to training transformer language models with up to 1.3B parameters and demonstrate comparable performance to regular neural network training. ReLoRA saves up to 5.5Gb of RAM per GPU and improves training speed by 9-40% depending on the model size and hardware setup. Our findings show the potential of parameter-efficient techniques for large-scale pre-training. Our code is available on GitHub[2].

## 1 Introduction

Over the past decade, the machine learning field has been dominated by the trend of training increasingly overparameterized networks or adopting the "stack more layers" approach [Krizhevsky et al., 2012, He et al., 2016, Kaplan et al., 2020]. The definition of a large network has evolved from models with 100 million [Simonyan and Zisserman, 2015, Radford et al., 2018] to hundreds of billions [Brown et al., 2020, Chowdhery et al., 2022] of parameters, which has made computational costs associated with training of such networks prohibitive to most of the research groups. Despite this, the necessity to train models which can have orders of magnitude more parameters than the training examples [Brown et al., 2020, Chowdhery et al., 2022, Fedus et al., 2022], is poorly understood theoretically [Jacot et al., 2018, Allen-Zhu et al., 2019, Zhang et al., 2021].

Alternative approaches to scaling, such as more compute-efficient scaling optima [Hoffmann et al., 2022], retrieval-augmented models [Khandelwal et al., 2020, Borgeaud et al., 2022], and the simple approach of training smaller models for longer [Touvron et al., 2023], have offered new trade-offs. However, they do not bring us closer to understanding why we need overparameterized models and rarely democratize the training of these models. For example, training RETRO [Borgeaud et al., 2022] requires a complex training setup and infrastructure capable of quickly searching over trillions of tokens, while training LLaMA-7B [Touvron et al., 2023] still requires hundreds of GPUs.

In contrast, approaches like zero-redundancy optimizers [Rajbhandari et al., 2020], 16-bit training [Micikevicius et al., 2018], 8-bit inference [Dettmers et al., 2022], and parameter-efficient fine-tuning (PEFT) [Lialin et al., 2023] have played a crucial role in making large models more accessible.

---

[*]Correspondance to vlad.lialin@gmail.com

[2]github.com/guitaricet/relora

Workshop on Advancing Neural Network Training at 37th Conference on Neural Information Processing Systems (WANT@NeurIPS 2023).

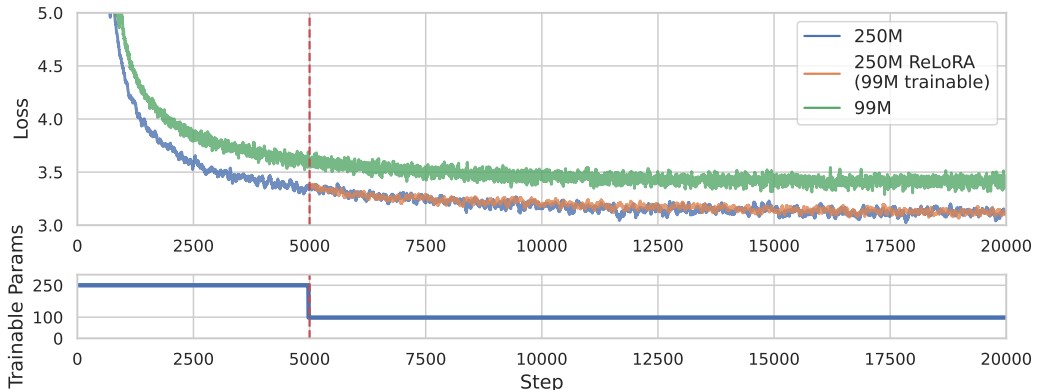

Figure 1: Training loss for 250M models. ReLoRA learns a high-rank network through a sequence of low-rank updates. It outperforms networks with the same trainable parameter count and achieves similar performance to training a full network at 100M+ scale. The efficiency of ReLoRA increases with the model size, making it a viable candidate for multi-billion-parameter training.

Specifically, PEFT methods have enabled fine-tuning of billion-scale language or diffusion models on consumer hardware. This raises the question: Can these approaches also benefit pre-training?

**Our Contribution** In this study, we introduce ReLoRA which uses individually low-rank updates that aggregate during the training process to train a high-rank network. We empirically demonstrate that ReLoRA performs a high-rank update and achieves performance similar to regular neural network training. The components of ReLoRA include initial full-rank training of the neural network (similar to Frankle et al. [2019]), LoRA training, restarts, a jagged learning rate schedule, and partial optimizer resets. We evaluate ReLoRA on transformer language models up to 1.3B parameters. Finally, we observe that the efficiency of ReLoRA increases with model size, making it a viable option for efficient training of multi-billion-parameter networks.

## 2 Method

We are interested in the rank of the sum of two matrices: $\text{rank}(A + B) \leq \text{rank}(A) + \text{rank}(B)$. We know that for a matrix $\mathbf{A}, \text{rank}(\mathbf{A}) < dim(\mathbf{A})$, there exists a $\mathbf{B}, \text{rank}(\mathbf{B}) < dim(\mathbf{B})$ such that sum of them has a higher rank than either $\mathbf{A}$ or $\mathbf{B}$.

We want to exploit this property to make a flexible parameter-efficient training method. We start with LoRA [Hu et al., 2022] which is a parameter-efficient fine-tuning method based on the idea of low-rank updates. LoRA can be applied to any linear operation parametrized through $W \in \mathbb{R}^{m \times n}$. Specifically, LoRA decomposes the weight update $\delta W$ into a rank-$r$ product $W_A W_B$ as shown in Equation 1, where $s \in \mathbb{R}$ is a fixed scaling factor usually equal to $\frac{1}{r}$.

$$\delta W = s W_A W_B$$
$$W_A \in \mathbb{R}^{\text{in} \times r}, W_B \in \mathbb{R}^{r \times \text{out}} \tag{1}$$

In practice, LoRA is usually implemented by adding new trainable parameters $W_A$ and $W_B$, which could be merged back into the original parameters after training. Thus, these implementations are restricted by the rank $r = \max_{W_A, W_B} \text{rank}(W_A W_B)$.

If we could restart LoRA, meaning we merge $W_A$ and $W_B$ during training and reset the values of these matrices, we could increase the total rank of the update. Doing this multiple times brings the total neural network update to:

$$\Delta W = \sum_{t=0}^{T_1} \delta W_t + \sum_{t=T_1}^{T_2} \delta W_t + \cdots + \sum_{t=T_{N-1}}^{T_N} \delta W_t = s W_A^1 W_B^1 + s W_A^2 W_B^2 + \cdots + s W_A^N W_B^N \tag{2}$$

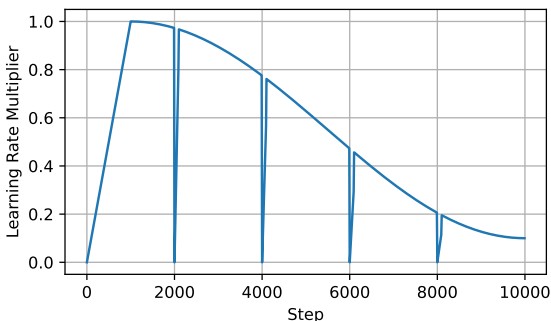

Figure 2: Jagged cosine scheduler used in ReLoRA. As a base for our scheduler we follow a standard cosine decay schedule as in Touvron et al. [2023]. On every optimizer reset, we set the learning rate to zero and perform a quick (50-100 steps) learning rate warm-up back to the cosine schedule.

However, implementing restarts is not trivial in practice and requires several modifications to the optimization procedure. Unlike plain stochastic gradient descent, Adam [Kingma and Ba, 2015] update is guided mainly by the first and second moments of the gradient accumulated over the previous steps. In practice, Adam's $\beta_1$ and $\beta_2$ are usually very high $0.9 - 0.999$. This means that after the merge-and-reinit, continuing to use old gradient moments for $W_A^2$ will guide it in the same direction as $W_A^1$ and optimize the same subspace.

To resolve this issue, ReLoRA performs a partial reset of the optimizer state during merge-and-reinit via magnitude pruning. To avoid loss diverging after an optimizer reset it also sets the learning rate to 0 with a subsequent warm-up (Figure 2). Our ablation studies (Table 6) show that both of these modifications are required to improve the performance over LoRA. Finally, in our experiments we found that in the case of training from scratch (random initialization) a short full-rank training is needed to "warm start" ReLoRA. All of this allows ReLoRA to achieve performance comparable to full-rank training, especially in large transformer networks, by only training a small set of parameters at a time. ReLoRA is described in Algorithm 1.

**Enhancing computational efficiency**     Unlike other low-rank training techniques [Schotthöfer et al., 2022, Sui et al., 2023, Kamalakara et al., 2022], ReLoRA follows the LoRA approach by maintaining the frozen weights of the original network and adding new trainable parameters. At first glance, this may appear computationally inefficient; however, the differentiation between frozen and trainable parameters plays a crucial role in parameter-efficient fine-tuning [Lialin et al., 2023].

By reducing the number of trainable parameters, ReLoRA significantly reduces the memory spent on the optimizer states and enables the utilization of larger batch sizes, maximizing hardware efficiency. Additionally, it reduces the bandwidth requirements in distributed setups, which are often the limiting factor in large-scale training. Furthermore, since the frozen parameters are not being updated between restarts, they can be kept in a low-precision quantized format [Dettmers et al., 2023], further reducing their memory and computational impact.

**Locally Low-Rank Training: Intuition**     Multiple studies suggest that neural network training is either completely low-rank or has multiple phrases with initially high-rank and subsequent low-rank training. For example, Aghajanyan et al. [2021] show that as the model becomes larger or when it is pre-trained for longer, the rank of the update needed to learn a downstream task reduces. Arora et al. [2019] finds that SGD is biased towards low-rank solutions. The existence of Lottery Tickets early in training [Frankle et al., 2019] also partially supports this hypothesis, since training a lottery ticket network could effectively be seen as a low-rank approximation to the regular training process. Our empirical analysis (Section 4) shows that pre-trained neural networks exhibit high-rank updates over long trajectories (Figure 4). However, for a sufficiently small trajectory, the training can be effectively approximated by a low-rank update. Given the above results, we speculate that neural network training is locally low-rank, which directly motivates ReLoRA.

**Algorithm 1** ReLoRA. $\theta$ is model parameters, $\hat{\theta}$ is model parameters with linear layers replaced with ReLoRA, $M$ and $V$ are Adam optimizer states, $\eta$ is learning rate, and $q$ is the reinit frequency.

---

**Require:** $\theta, M, V, q, \eta$
 1: **for** t **in** warm start steps **do**
 2:     Update $\theta, M, V, \eta$ {Regular training for warm start}
 3: **end for**
 4: **for** layer in model layers **do**
 5:     **if** layer **is** linear **then**
 6:         layer $\leftarrow$ ReLoRA$(W^i, W_A^i, W_B^i)$
 7:         Freeze $W^i$
 8:     **end if**
 9: **end for**
10: **for** t in training steps **do**
11:     Update $\hat{\theta}, M, V$ {Training step with ReLoRA}
12:     **if** MOD$(t, q) = 0$ **then**
13:         **for** l in model layers **do**
14:             **if** l **is** linear **then**
15:                 $W^i \leftarrow (W^i + sW_A^i W_B^i)$
16:                 $W_A^i \leftarrow$ kaiming_init$(W_A^i)$; $W_B^i \leftarrow 0$
17:                 $M_{W_A^i} \leftarrow$ prune$(M_{W_A^i})$; $V_{W_A^i} \leftarrow$ prune$(V_{W_A^i})$
18:             **end if**
19:         **end for**
20:         Start $\eta$ warmup
21:     **end if**
22: **end for**
23: **return** $\theta$

---

## 3    Experiments

To evaluate the effectiveness of ReLoRA, we apply it to train a transformer language model on the C4 dataset [Raffel et al., 2020] using various model sizes: 60M, 130M, 250M, 350M, and 1.3B.

In all experiments we train without data repetition (single epoch) on at least compute-optimal amount of data, estimated using Chinchilla Scaling Laws [Hoffmann et al., 2022].

**Architecture and training hyperparameters**    Our architecture is based on transformer [Vaswani et al., 2017] and closely resembles LLaMA [Touvron et al., 2023]. Namely, we use pre-normalization, RMSNorm [Zhang and Sennrich, 2019], SwiGLU activations [Shazeer, 2020], $\frac{8}{3}h$ fully-connected hidden state size [Touvron et al., 2023], and rotary embeddings [Su et al., 2021]. We select the number of pre-training tokens based on the Chinchilla scaling laws [Hoffmann et al., 2022]. Architecture and training hyperparameters are presented in Table 1.

For all LoRA and ReLoRA experiments, we use rank $r = 128$ as our initial experiments showed it to have the best perplexity/memory trade-off. You can find additional recommendations on ReLoRA hyperparameter selection in Appendix A. We perform additional experiments comparing different

| Params | Hidden | Heads | Layers | Learning rate | Batch size | Seq. len. | Data amount |
|--------|--------|-------|--------|---------------|------------|-----------|-------------|
| 60M    | 512    | 8     | 8      | 1e-3          | 122K       | 256       | 1.2B        |
| 130M   | 768    | 12    | 12     | 1e-3          | 154K       | 256       | 2.6B        |
| 250M   | 768    | 16    | 24     | 5e-4          | 590K       | 512       | 6.8B        |
| 350M   | 1024   | 16    | 24     | 5e-4          | 590K       | 512       | 6.8B        |
| 1.3B   | 2048   | 24    | 32     | 4e-4          | 786K       | 512       | 23.1B       |

Table 1: Hyperparameters of the language models trained in this study. Batch size and data amount are specified in tokens.

|                   | 60M           | 130M          | 250M          | 350M          | 1.3B          |
| ----------------- | ------------- | ------------- | ------------- | ------------- | ------------- |
| Full training     | 33.81 (60M)   | 23.65 (130M)  | 22.39 (250M)  | 18.66 (350M)  | 16.83 (250M)  |
| Control           | 36.52 (43M)   | 27.30 (72M)   | 25.43 (99M)   | 23.65 (130M)  | 21.73 (250M)  |
| LoRA              | 47.44 (43M)   | 34.17 (72M)   | 36.60 (99M)   | 57.11 (125M)  | -             |
| LoRA + Warm Start | 34.73 (43M)   | 25.46 (72M)   | 22.86 (99M)   | 19.73 (125M)  | 18.23 (250M)  |
| ReLoRA            | **34.46** (43M) | **25.04** (72M) | **22.48** (99M) | **19.32** (125M) | **17.27** (250M) |
| Training tokens   | 1.2B          | 2.6B          | 6.8B          | 6.8B          | 23.1B         |

Table 2: Language model perplexity when trained using each of the above methods. Number of trainable parameters for each model in (brackets). Control baseline is full-rank training a model with the same total number of parameters as the number of trainable parameters in low-rank training.

|                        | CoLA  | STS-B | MRPC  | RTE   | SST2  | MNLI  | QNLI  | QQP   | Avg   |
| ---------------------- | ----- | ----- | ----- | ----- | ----- | ----- | ----- | ----- | ----- |
| Full-rank pretrained   | 35.43 | 83.85 | 76.96 | 64.26 | 88.99 | 70.98 | 83.38 | 84.49 | 73.54 |
| Not pretrained         | 7.59  | 22.73 | 67.00 | 51.15 | 82.61 | 60.04 | 67.92 | 78.40 | 54.68 |
| ReLoRA                 | 31.07 | 83.33 | 78.43 | 60.65 | 89.45 | 72.27 | 83.93 | 86.01 | 73.14 |

Table 3: Applying ReLoRA to fine-tune 350M models pre-trained full-rank and using ReLoRA. We observe minimal differences between the models.

rank choices for the 1.3B model in Section 4.1. We use bfloat16 for all floating point operations and FlashAttention [Dao et al., 2022] for effective attention computation.

**ReLoRA and baselines setup**   In our experiments, ReLoRA replaces all attention and fully-connected network parameters, while updating the embeddings and normalization layers full-rank. Since ReLoRA-wrapped models have fewer trainable parameters than full-rank training, we include a Control baseline, which is a full-rank transformer with the same number of trainable parameters as ReLoRA.

We initialize ReLoRA from a checkpoint of full-rank training at 5,000 update steps and reset it every 5,000 steps thereafter, 3 times in total till we reach 20K steps. After each reset, 99% of the optimizer state is pruned based on magnitude, and the loss is warmed up for the next 100 iterations. ReLoRA parameters are reinitialized following LoRA best practices, Kaiming initialization [He et al., 2015] for $A$-matrix, and zeros for $B$-matrix.

**Scaling up to 1.3B**   After initial results at 130M and 350M model sizes, we applied ReLoRA to train a 1.3B parameter language model. As a baseline, we pre-trained a 1.3B model from scratch on 23B tokens. We performed multiple ReLoRA runs starting from 2K, 5K, and 10K checkpoints. In most of the experiments, we continued using $r = 128$ and our additional experiments show minimal difference between rank 128 and 512 (hidden size is 2048). Section 4.1 describes these experiments in detail.

## 4   Results

**Parameter-efficient pre-training**   Our results are presented in Table 2 and Figure 1. ReLoRA significantly outperforms LoRA training demonstrating the effectiveness of our proposed modifications (ablated in Section 6). Additional pre-training loss figures are available in Appendix C.

Furthermore, ReLoRA achieves similar performance to full-rank training in both upstream and downstream tasks (Table 3).[3]

**High-rank training through low-rank updates**   To determine whether ReLoRA performs a higher rank update than LoRA, we plot the singular value spectrum of the learned update to the warm-start

---

[3]Note that the absolute values of GLUE results are expected to be quite far from state-of-the-art, because our models were pre-trained on roughly 20 times less data than T5 or BERT.

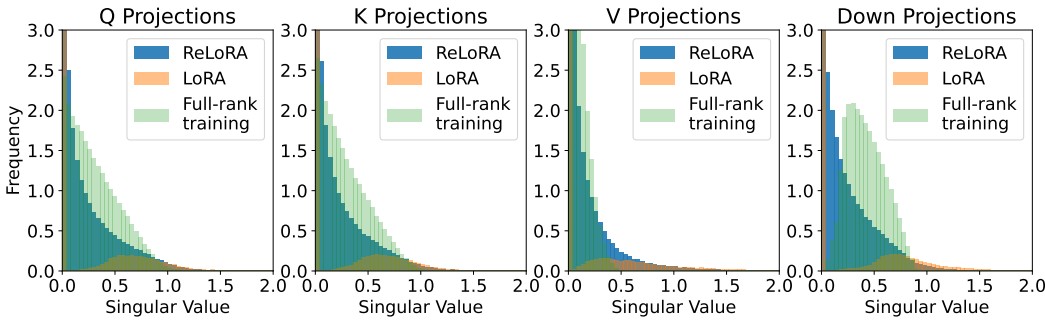

Figure 3: Singular values spectra of the weight difference between ReLoRA and LoRA at 5,000 iterations (warm start) and 20,000 iterations. ReLoRA exhibits a closer resemblance to full-rank training than to LoRA, indicating its effectiveness in approximating full-rank behavior. 350M models.

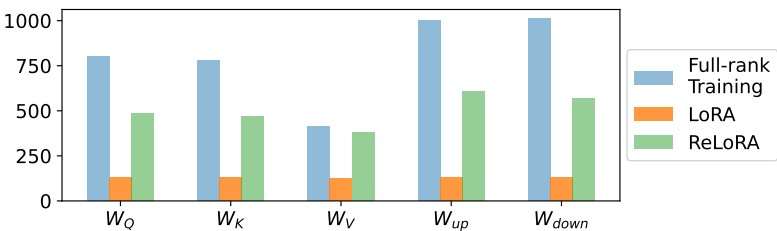

Figure 4: The number of singular values >0.1 in weight matrices of the learned update. 350M models.

weights. Specifically, the difference between warm-start weights and the final weights for ReLoRA, LoRA, and full-rank trained models. Figure 3 illustrates significant qualitative differences between LoRA and ReLoRA for the singular values of $\Delta W_Q$, $\Delta W_K$, $\Delta W_V$, and $\Delta W_{down}$. While most of the singular values for LoRA are zero (Figure 4) with a noticeable number of exceptionally high values above 1.5, ReLoRA exhibits a higher distribution mass between 0.1 and 1.0, reminiscent of full-rank training.

Additionally, we computed the number of singular values less than 0.1 for LoRA, ReLoRA, and full-rank training. Our results (Figure 4) show that ReLoRA has a much smaller number of near-zero singular values than LoRA, closer to full-rank training. This observation emphasizes the significance of high-rank updates and demonstrates that ReLoRA does accomplish a high-rank update by performing multiple low-rank updates. We also perform ReLoRA component ablation (Table 6) and discuss it in Section 6.

## 4.1 Scaling up to 1.3B

Our best run at this model size starts after a 10K step warm start (33% of the total update steps). We train ReLoRA with rank $r = 128$, learning rate 5e-4, 100 steps lr warmup, and 50 steps restarts warmup. The results are presented in the Figure 5 and Table 4. ReLoRA clearly outperforms LoRA throughout the training with the gap between the methods increasing from 0.56 at 15K steps to 0.96 at 30K steps. At the end of the training, ReLoRA is able to reach a perplexity of 17.24, only 0.44 higher than full-rank training. You can find additional recommendations on ReLoRA hyperparameter selection in Appendix A.

**Varying ReLoRA rank**  In this experiment we wanted to evaluate if $r = 128$ is still applicable to the model of this size (hidden size $2048$) or if it needs to be increased. To do that, we used an early checkpoint for the warm start (5K out of 30K steps). This was beneficial for the comparison, as at this point loss changes quickly which makes any differences in training dynamics more evident. We train these models for additional 10K iterations. Unexpectedly, we found very little difference between ranks 128 (ppl. 19.16) and 512 (ppl. 19.00).

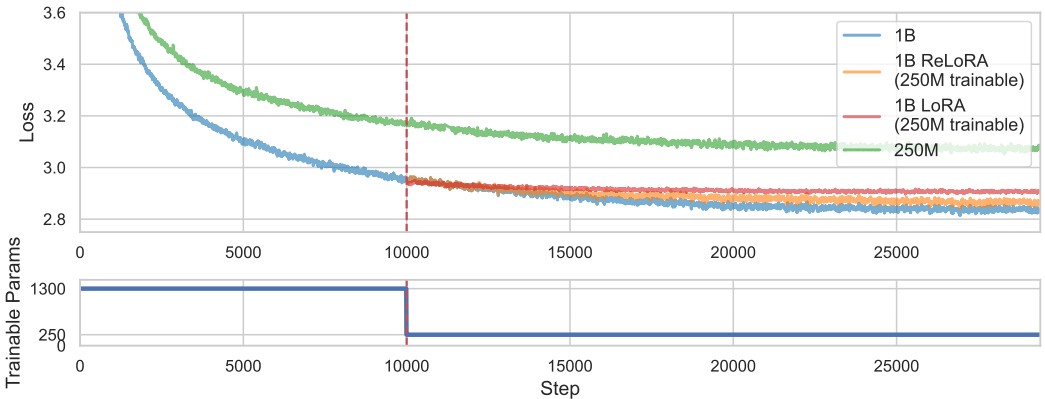

Figure 5: Training loss at 1.3B scale and the associated baselines. ReLoRA outperforms LoRA throughout training and the gap between the methods increases over time.

|  | 1.3B @15K steps | 1.3B @20K steps | 1.3B @30K steps |
|---|---|---|---|
| Full training | 17.67 (250M) | 17.00 (250M) | 16.83 (250M) |
| Control | 22.67 (250M) | 22.00 (250M) | 21.73 (250M) |
| LoRA + Warm Start | **18.50** (250M) | **18.38** (250M) | **18.23** (250M) |
| ReLoRA | **17.94** (250M) | **17.64** (250M) | **17.27** (250M) |
| Training tokens (billions) | 11.8 | 15.7 | 23.1 |

Table 4: Results at 1.3B scale. Number of trainable parameters for each model in (brackets).

**Negative results: Online ReLoRA** Intuitively, more frequent ReLoRA resets can lead to better performance, as they, in principle, can learn a higher rank update. Usually, for every ReLoRA reset, we would also perform an optimizer reset and learning rate scheduler re-warmup (Section 1). However, in our experiments we observed that very high ReLoRA reset rates lead to worse performance.

|  | 250M (@15k steps) | 1.3B (@25k steps) |
|---|---|---|
| ReLoRA | 27.66 | 17.36 |
| Online ReLoRA | 29.31 | 17.80 |

Table 5: Online ReLoRA.

Online ReLoRA resolves this issue quite elegantly – it merges LoRA parameters very frequently (e.g., every 100 iterations) while keeping the optimizer reset rate at 2-5K iterations. Unexpectedly, we found that it performs worse than regular ReLoRA at both 250M and 1.3B scales (Table 5).

**ReLoRA Training Speedup** Training ReLoRA took 440 A100-hours, saving 56 A100-hours compared to full-rank training. A part of the speedup was due to the ability to use two times larger microbatch size. When training with the same microbatch size, ReLoRA improved RAM consumption from 27.8Gb to 22.3Gb saving 5.5Gb of GPU RAM. Overall, in the 8xA100 setup, combining the warm start and ReLoRA training time, 1.3B-ReLoRA took 86 hours (wall clock) to train compared to 93.5 hours to train 1.3 model full-rank on the same amount of data. This yields a relative speed improvement of 9%.

We additionally observed that ReLoRA speedup is significantly hardware-dependent (Table 7). In our early experiments on 2xRTX3090, we estimated the speedup of 42%. In a more practical, but still relatively budget setup of 6xA6000 Ada, we estimated 152 hours of wall-clock training time for the 1B full-rank model and 119 hours for the ReLoRA model with 33% warm start. This saves 33 hours yielding a relative speedup of 21%. We attribute the difference to the GPU memory speed. ReLoRA can more effectively utilize low-bandwidth memory as it has less trainable parameters.

| Restarts | Optimizer Reset | Jagged Schedule | Warm Start | Perplexity ($\downarrow$) |
|---|---|---|---|---|
| ✗ | ✗ | ✗ | ✗ | 34.17 |
| ✓ | ✗ | ✗ | ✗ | 34.25 |
| ✓ | ✓ | ✗ | ✗ | *(diverged)* |
| ✓ | ✗ | ✓ | ✗ | 34.29 |
| ✓ | ✓ | ✓ | ✗ | 29.77 |
| ✗ | ✗ | ✗ | ✓ | 25.46 |
| ✓ | ✓ | ✓ | ✓ | 25.04 |
| Regular training | | | | 23.65 |

Table 6: Ablation studies of ReLoRA (130M models). Restarts and warm starts are essential for good performance. Restarts and optimizer resets without a jagged schedule causes the model to diverge.

| | 8xA100 | 6xA6000 (Ada) | 2x3090 |
|---|---|---|---|
| Full-rank throughput | 137 ex/sec | 84 ex/sec | 8.8 ex/sec |
| ReLoRA throughput | 157 ex/sec | 124 ex/sec | 17.8 ex/sec |
| Immediate speedup | 15% | 48% | 102% |
| Warm-start adjusted ReLoRA throughput | 149 ex/sec | 111 ex/sec | 14.8 ex/sec |
| Total speedup | 9% | 32% | 51% |

Table 7: Performance metrics in different hardware configurations. Warm start adjustment assumes 33% of full-rank training before switching to ReLoRA.

## 4.2 Ablation studies

We conduct ablation studies on all four crucial components of ReLoRA: restarts, jagged schedule, optimizer resets, and warm starts, utilizing the 130M-sized model. The results are presented in Table 6. In this section, we will focus on and analyze certain combinations of these components.

**LoRA**  ReLoRA, without the aforementioned components, is essentially equivalent to training a low-rank network parameterized by LoRA. This approach yields remarkably high perplexity, indicating that a simple matrix decomposition has significantly different training dynamics from full-rank training.

**Adding restarts and optimizer resets**  ReLoRA, without a jagged schedule and optimizer reset, performs similarly to LoRA because old optimizer states force the newly initialized parameters into the same subspace as the prior weights, limiting the model's capacity. However, doing a naive optimizer reset with ReLoRA causes the model to diverge. A jagged schedule helps to stabilize training and has a positive impact on the mixture. In our initial experiments, we also observed that a combination of partial optimizer reset and jagged scheduler allows for a quicker warm-up, as low as 50 steps, instead of hundreds of steps required when the optimizer is initialized from scratch.

**Warm start**  The warm start shows the most significant improvement, dropping perplexity by almost 10 points. To investigate whether post-warmup training contributes to the loss, we measured the perplexity of the warmed-up network, which equals $27.03$. It outperforms all low-rank methods except for our final ReLoRA recipe but still demonstrates a significant difference from the final network. This demonstrates the importance of early training, similar to the concept of the lottery ticket hypothesis with rewinding [Frankle et al., 2019]. In our experiments, unless specified otherwise, we performed warm start for about $1/4$ of the total training updates.

## 5  Related work

**Scaling versus Efficiency**  The relationship between overparametrization and neural network trainability and generalization has been extensively studied [Zhang et al., 2017, Belkin et al., 2018,

Frankle and Carbin, 2019, Nakkiran et al., 2019, Singh et al., 2021], yet it remains a mystery [Zhang et al., 2021].

Moreover, scaling laws [Kaplan et al., 2020, Ghorbani et al., 2021, Hoffmann et al., 2022] demonstrate a simple and strong power-law dependence between network size and its performance across a variety of modalities. This finding not only supports overparametrization but also encourages the training of extraordinarily resource-intensive neural networks [Brown et al., 2020, Chowdhery et al., 2022, Fedus et al., 2022]. Nonetheless, the Lottery Ticket Hypothesis [Frankle et al., 2019] suggests that overparametrization could, in principle, be minimized.

**Parameter-efficient fine-tuning**    Aghajanyan et al. [2021] found that pre-training reduces the amount of change to the network required to learn a new task through fine-tuning. I.e., larger networks or networks pre-trained on more data require smaller modifications in terms of the rank of the range to learn a new task. This explains the success of parameter-efficient fine-tuning methods [Lialin et al., 2023] and has also motivated the development of low-rank fine-tuning methods such as LoRA [Hu et al., 2022] and Compacter [mahabadi et al., 2021].

**Low-rank neural network training**    Training low-rank representations has been explored in the context of CNN compression, regularization, and efficient training [Idelbayev and Carreira-Perpinan, 2020, Jaderberg et al., 2014, Sui et al., 2023, Schotthöfer et al., 2022, Lin et al., 2020, Yuan et al., 2021, Zhao et al., 2023]. However, most of these methods are either specific to CNNs, do not scale well, or have not been evaluated on large transformers [Vaswani et al., 2017] with hundreds of millions of parameters, which can benefit greatly from efficient training. While transformers have been shown to have a low-rank internal dimensionality and representations [Aghajanyan et al., 2021, Wang et al., 2020], the study by Bhojanapalli et al. [2020] demonstrated that the low rank of key and query projections in multi-head attention bottlenecks the performance of transformers. Our own experiments (Section 6) also demonstrate that low-rank transformers perform significantly worse compared to the full-rank baseline and ReLoRA.

## 6    Conclusion

In this paper, we demonstrate that parameter-efficient fine-tuning methods can be adapted for pre-training large language models. We first examined the limitations of a low-rank matrix factorization (LoRA) approach and observed that it struggles to effectively train high-performing transformer models. To address this issue, we proposed ReLoRA, which leverages the rank of sum property to train a high-rank network through multiple low-rank updates. Similar to the lottery ticket hypothesis with rewinding, ReLoRA employs a full-rank training warm start before transitioning to ReLoRA. During training, ReLoRA periodically merges its parameters into the main parameters of the network, performs optimizer reset and learning rate re-warmup.

We demonstrated that ReLoRA consistently outperforms LoRA for training large transformer models. Our largest experiment demonstrated $9\%$ wall-clock time reduction in 8xA100 setup and much larger $(20 - 40\%)$ speed improvements on cheaper hardware. Further, our results show similar performance to regular training making ReLoRA a promising candidate for improving the efficiency of large model training. Our further studies will focus on improving ReLoRA performance, efficiency, applying it to larger models and applying it to continued pre-training of existing large language models.

## Acknowledgments and Disclosure of Funding

This paper has been a journey and we are sincerely grateful to everyone who supported us. We would like to express our gratitude to Stability.ai, Eleuther.ai, and the Google Cloud for Research Program for providing computational resources essential for this research.

Eric Lehman and Artem Krivosheev, thank you for supporting this project from the very beginning.

Special thanks to Jason Phang, Hailey Schoelkopf, Enrico Shippole, and Stella Biderman for their technical advice and assistance with computational resources. Our experiments at billion-parameter scale wouldn't be possible without your support.

This work was funded in part by an Amazon Alexa AI research award to Anna Rumshisky.

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

# A    A Practical guide to ReLoRA

In this section, we wanted to answer most common questions on hyperparameter selection. Especially how to select ReLoRA-specific hyperparameters to reliably get better performance than LoRA. In all of our experiments, we applied LoRA/ReLoRA to all of the linear layers in the model: kqv-projection layers, FFN layers and other projections, except for logits and embeddings.

We observed that $r \in \{64, 128\}$ works well for all of the networks, up to 1B. One small, but important hyperparameter change from full-rank training to ReLoRA-training that was crucial for the performance was increased learning rate. ReLoRA (and LoRA) requires $1.5 - 2$ times larger learning rate than regular training/fine-tuning to achieve similar performance.

When taking about ReLoRA-specific hyperparameters, we did not observe significant dependence on optimizer pruning percentage as long as it's larger than 90%. Larger pruning rates can lead to slightly better performance at the cost of possible loss instabilities during the reset. We tested several ReLoRA reset rates with 350M and 1.3B models and found that 2K iterations reset rate performed consistently well in both pre-training and fine-tuning experiments and always led to better performance than no resets. In general, we observed good results with reset rates 2K-5K.

# B    ReLoRA for fine-tuning

We apply ReLoRA to fine-tune T5-base (220M parameters) and T5-large (770M parameters) on the GLUE benchmark. We use the same type of learning rate scheduler as in ReLoRA pre-training and prune 90% of the low magnitude optimizer states during each LoRA merge-and-reinit (restart). The batch size is equal to 128 examples and the learning rate is tuned (from 1e-4 to 5e-4) on each model and dataset combination. We perform additional ReLoRA ablation studies using the T5-Large model and QNLI dataset. Specifically, we explore different ReLoRA ranks, optimizer state pruning rates, and the total number of ReLoRA resets.

| Method | SST-2 | MNLI | QNLI | QQP | RTE | STS-B | MRPC | CoLA | Avg |
|---|---|---|---|---|---|---|---|---|---|
| Adapters[†] | 94.2 | 86.4 | 93.1 | 88.9 | 75.1 | 91.1 | 88.9 | 64.4 | 85.3 |
| Prompt Tuning[†] | 90.3 | 82.5 | 92.5 | 88.5 | 59.5 | 90.1 | 74.6 | 0.0 | 72.2 |
| Ladder Side Tuning[†] | 94.1 | 85.6 | 93.3 | 88.8 | 71.9 | 90.7 | 90.4 | 58.1 | 84.1 |
| Compacter[*] | 93.9 | 86.1 | 92.9 | 90.4 | 76.3 | 91.0 | 91.5 | 64.4 | 85.8 |
| KronA[*] | 94.3 | 86.3 | 93.2 | 90.6 | 77.7 | 91.3 | 92.5 | 63.3 | 86.1 |
| Full fine-tuning[*] | 93.6 | 86.2 | 92.8 | 91.7 | 74.8 | 90.1 | 92.7 | 63.4 | 85.7 |
| LoRA | 93.92 | 86.12 | 91.95 | 90.62 | 78.34 | 89.96 | 90.52 | 60.04 | 85.18 |
| ReLoRA | 94.15 | 85.96 | 91.68 | 87.2 | 77.74 | 89.88 | 90.03 | 59.92 | 84.57 |
| Full fine-tuning (T5-L) | 94.7 | 89.1 | 91.6 | 89.9 | 78.9 | 90.6 | 88.9 | 57.0 | 85.0 |
| LoRA (T5-L) | 95.59 | 89.44 | 93.98 | 91.44 | 85.92 | 90.89 | 92.90 | 63.77 | 87.99 |
| ReLoRA (T5-L) | 95.7 | 89.06 | 93.68 | 91.04 | 84.72 | 90.53 | 90.57 | 61.72 | 87.47 |

Table 8: ReLoRA for fine-tuning does not outperform LoRA. Results with [†] and [*] are T5-base results from Sung et al. [2022] and Edalati et al. [2022] respectively.

**ReLoRA fine-tuning ablations**    Table 9 shows the results of varying ReLoRA hyperparameters. A rank of 64 seems to provide the best performance. The results indicate that the model's performance remains largely unaffected even when 99% of the optimizer states are reset. Our analysis of the jagged cosine learning rate scheduler's impact on classification accuracy in the QNLI dataset suggests that two resets are adequate (reset rate 4000).

# C    Learning curves of models pre-trained in the study

In this section we present additional training loss plots for all of the models from Table 2. 60M: Figure 6, 130M: Figure 7, 250M: Figure 8, 350M: Figure 9, 1.3B: Figure 10.

| Rank | Acc. | Pruning | Acc. | Reset rate | #resets | Acc. |
|---|---|---|---|---|---|---|
| 16 | 94.05 | 85% | 94.51 | 6000 | 1 | 94.38 |
| 32 | 94.16 | 92% | 94.33 | 4000 | 2 | **94.73** |
| 64 | **94.55** | 95% | 94.31 | 2000 | 5 | 94.34 |
| 128 | 94.44 | 99% | **94.56** | 1000 | 11 | 94.33 |

Table 9: ReLoRA fine-tuning ablations. We apply ReLoRA to fine-tune T5-large on the QNLI dataset and vary LoRA rank ($r$), optimizer state pruning percentage, and reset frequency of ReLoRA. Reset rate means the number of iterations between ReLoRA resets.

## D   Ranks of 130M models

Figures 11 and 12 show spectral properties for 130M model.

## E   Smaller warm start period

Table 2 demonstrates that ReLoRA consistently outperforms the warmed-started LoRA baseline. To provide a more contrasting example, we performed additional pre-training experiments starting from just 2K warm-started network. Figure 13 shows a significant performance gain with ReLoRA over LoRA by 1.4 ppl points (ppl 23.64 vs 25.08). While the absolute performance of ReLoRA is lower compared to full-rank training in this context, these experiments validate our initial hypothesis that LoRA restarts positively impact performance.

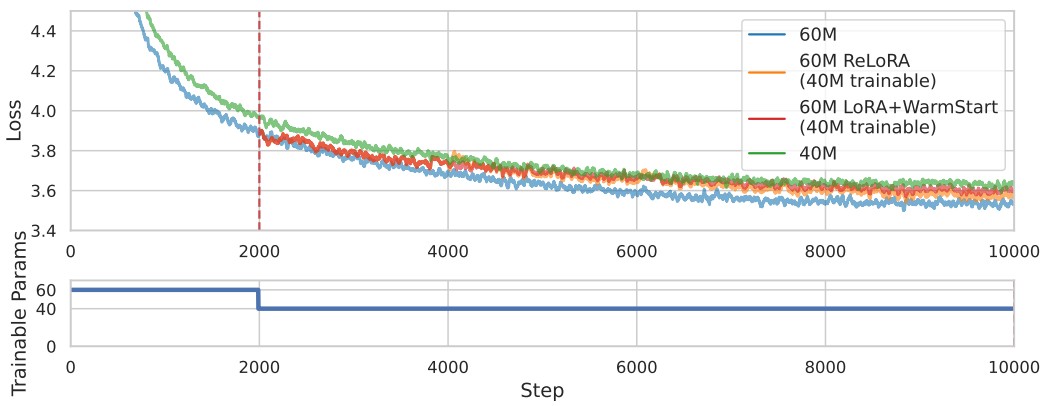

Figure 6: 60M experiments training loss

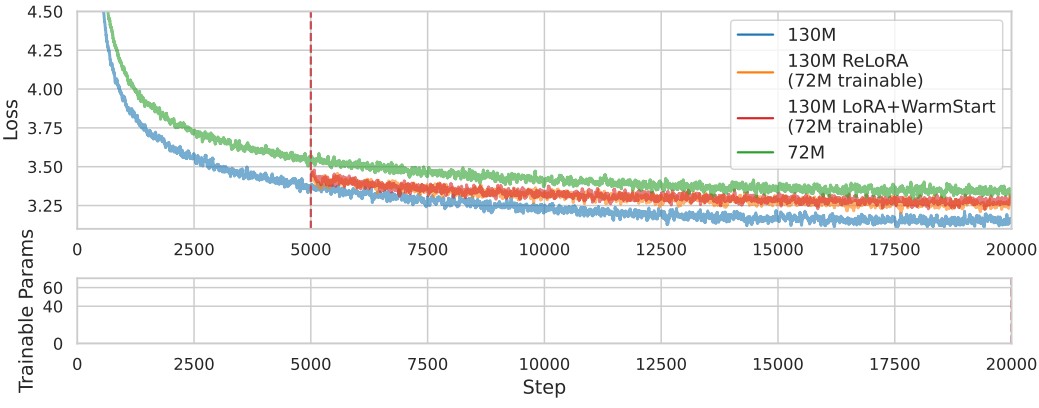

Figure 7: 130M experiments training loss

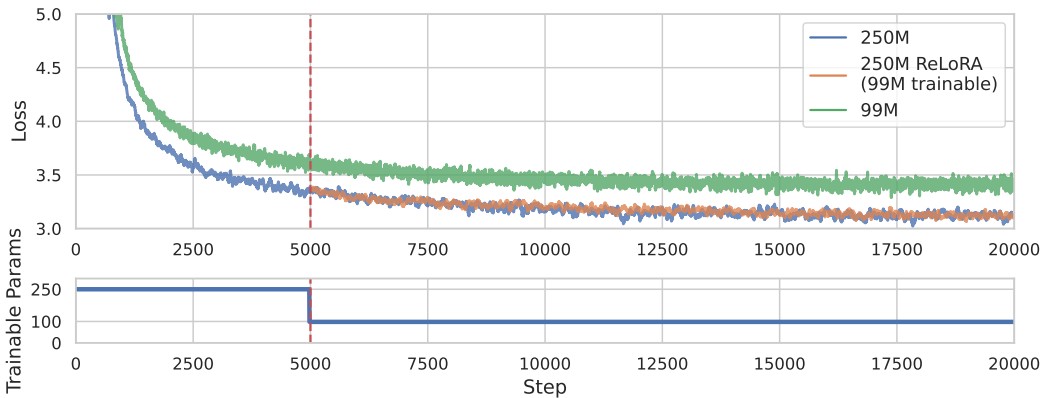

Figure 8: 250M experiments training loss

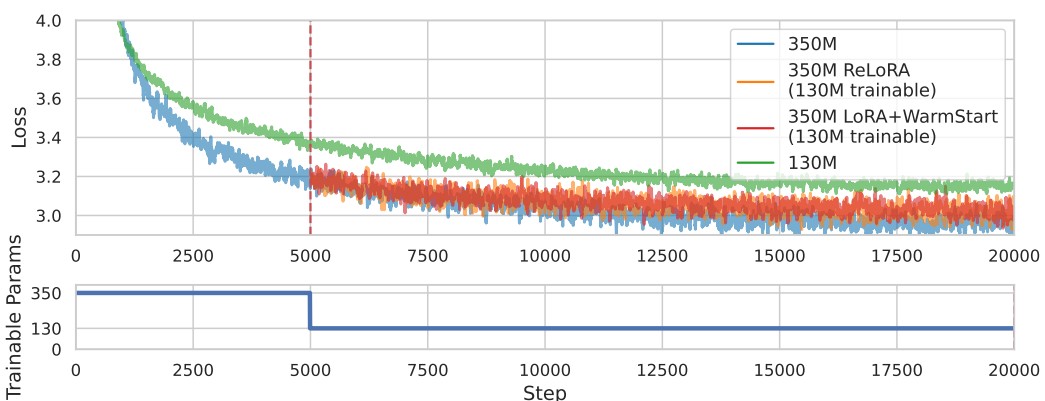

Figure 9: 350M experiments training loss

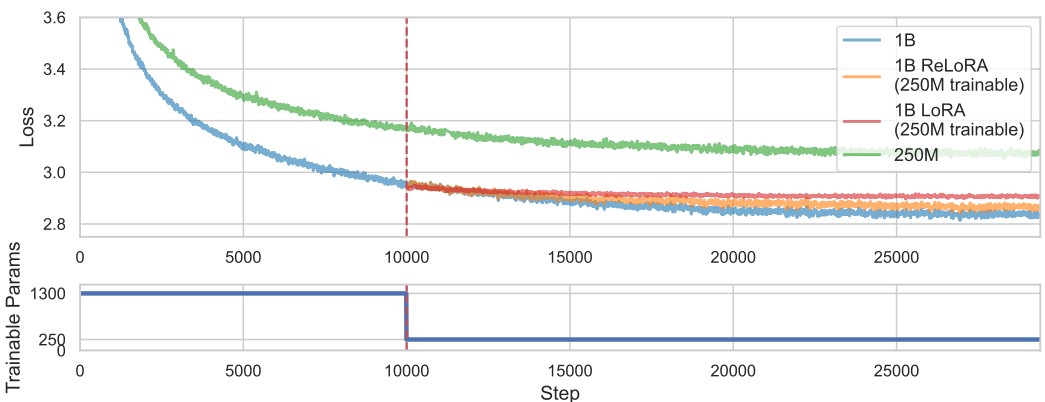

Figure 10: 1.3B experiments training loss

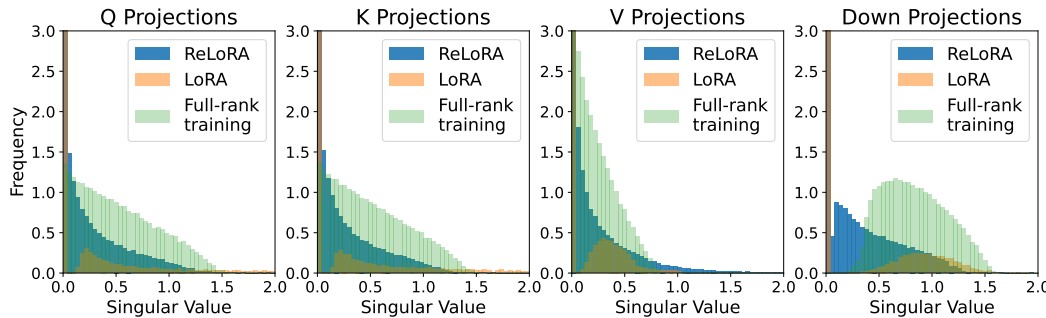

Figure 11: Singular values spectra of the weight difference between ReLoRA and LoRA at 5,000 iterations (warm start) and 20,000 iterations. ReLoRA exhibits a closer resemblance to full-rank training than to LoRA, indicating its effectiveness in approximating full-rank behavior. 130M models.

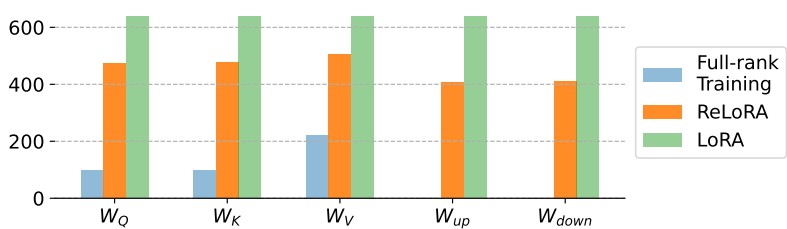

Figure 12: The number of singular values $< 0.1$ in attention and FCN matrices of the learned update. 130M models.

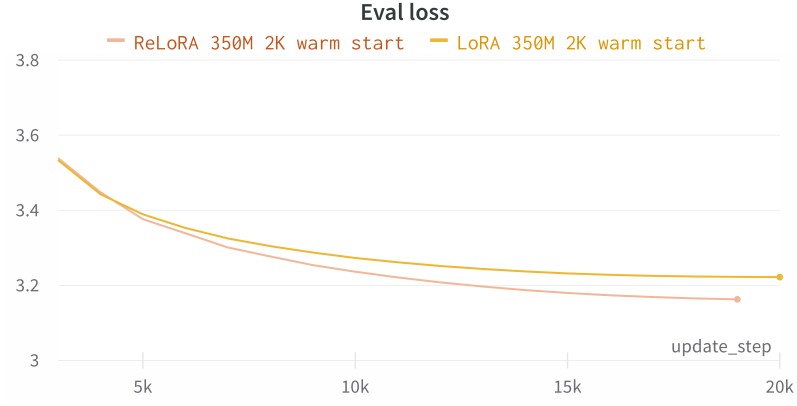

Figure 13: ReLoRA significantly outperforms LoRA when started from an early (2K steps) checkpoint.

