# OpenReview forum: "ReLoRA: High-Rank Training Through Low-Rank Updates"
_NeurIPS.cc/2023/Workshop/WANT — WANT@NeurIPS 2023 Poster_

### Official Review · Reviewer_HDtH · 2023-10-22
**A solid paper with extensive experiments**

**Confidence:** 4

**Review:**

The paper "ReLoRA: High-Rank Training Through Low-Rank Updates" proposes a novel method for training large neural networks called ReLoRA. The authors address the issue of large neural networks efficient training. The key idea behind ReLoRA is to utilize iterative low-rank updates to train high-rank networks. By applying this technique to transformer language models with up to 1.3B parameters, the authors demonstrate comparable performance to regular neural network training. Authors provide extensive experiments that prove the efficiency of the proposed method and show how different components of the framework affect its performance.

Overall, the findings of this paper highlight the potential of parameter-efficient techniques like ReLoRA for large-scale training. The proposed method offers a promising approach to training large neural networks while mitigating the associated costs and resource requirements.

Based on the significance of the contributions and the experimental results presented, I recommend accepting this paper for publication.

---

### Official Review · Reviewer_uFbN · 2023-10-24
**A lot of engineering efforts are put together to make ReLoRA work (like any other LLM training paper) but the story is a bit confusing.**

**Confidence:** 4

**Review:**

This paper introduces a new low rank training recipe called ReLoRA that utilizes LoRA, restart with warm ups and pruning of the state dictionary. The author's did a really good job in combining multiple techniques to make this recipe work. The paper is reasonably well written and easy to follow.

The major strong points in this paper are
1) The training methodology suggested in this paper is novel.
2) A lot of efforts are made to make ReLoRA work and also provide good insights.
3) The results and the speedups shown in the paper is very interesting and has the potential to improve pre-training.

Some key areas of improvements.
1) The paper started the story with training LLMs i.e. pre-training but at multiple occasions took a de tour into finetuning. The fine-tuning results are weak as the pre-training is conducted with lesser tokens than conventional T5 models. The finetuning baselines should be a trained T5 (original), LoRA, ReLORA and other baselines shown.
2) This paper could also focus only on pre-training because that's the main focus as highlighted in abstract and intro. Finetuning is just diluting the story.
3) The training recipe suggested by the authors requires a lot of other ingredients and it is not compatible with any conventional training.
4) The pre-training  results shown in Table 2 and Table 6 is not representative as one cannot see the training curve it is not clear how much the model has converged or if ReLoRA is beating conventional training in early training or also in the final phase.
5) The models are trained with less tokens and there is no information provided about data repetition (not at all or with replacement).
6) The pre-training PPL (loss) looks too high since these models are new there is no baseline provided about the minimal PPL these models can achieve with full c4 data. Therefore it is difficult to gauge how good the proposed recipe is compared to conventional training.
7) The paper says we use model similar to LLamA (line 95) but then at another place says initial results with BERT-BASE and BERT Large (line 115). Then in table 3 it shows results about finetuning encoder-decoder models (T5). I believe the evaluation should be limited to a specific set of models. This way of evaluation is a bit confusing to me.

Overall I believe this paper needs some rework on the evaluation side to makes the results more promising.

---

### Official Review · Reviewer_jQsG · 2023-10-25
**The manuscript proposes a training procedure which incorporates low-rank adapters with scheduled resets, optimizer states pruning, warm start and a jagged learning rate schedule. Although some experiments do not show significant superiority over LoRA+Warm Start training regime, results for pre-training larger Llama architectures(1.3B) are convincing.**

**Confidence:** 5

**Review:**

# Quality
## Pros
- The problem is formulated well and the algorithm description is clear and comprehensive.
- Authors have devised a specific jagged learning scheduler to compensate for optimizer states resets.
- The results on 1.3B Llama model pre-training and T5-Base fine-tuning are convincing of RELoRA's effectiveness.
- Exhaustive ablation study on RELoRA hyper-parameters (rank, % of pruning, reset rate) for RELoRA fine-tuning. (However, such ablation study were lacking for pre-training experiments.)

## Cons
- Results for T5-Large fine-tuning. No explanation on why the performance gap between RELoRA and LoRA + Warm Start on T5 fine-tuning task is diminishing with the model growth.
- It is doubtful whether the results for pre-training smaller Llama architectures are significantly better than that of LoRA + Warm Start pipeline, especially considering the used metric (perplexity).
- None of the results in the paper have std or confidence region provided which makes the doubt of significant gap between two methods even more prominent.

# Clarity
The pipeline and experiments description is clear, with some minor exceptions in results section (particularly, some graphs and tables). For example, Figure 3 depicts singular values spectra. It is not clear whether the singular values from both 5_000 and 20_000 iterations were mixed together for this evaluation or not. No std or confidence region for any of the results depicted in the paper.
The are few occasional typos throughout the manuscript. (i.e. line 181).

# Originality and Significance
The idea of scheduled reset of low-rank adapters is of limited novelty, however, optimizer state resets and jagged learning schedule that compensates for these resets together form a novel pipeline which can accelerate (or even make accessible) large LLM pre-training (as proven by Llama 1.3B experiment).

---

### Meta-Review · Area_Chair_YNCV · 2023-10-27

**Recommendation:** Accept (Poster)
**Confidence:** 4

**Metareview:**

This manuscript considers the parameter-efficient training problem and extends LoRA from the aspect of rank by incorporating three techniques. Topicwise this paper is also very interesting and relevant to the workshop.

Though the manuscript has some limitations, the insights might inspire interesting future research, for both theory and practice.

---

### Decision · Program_Chairs · 2023-10-28

**Decision:**

Accept (Poster)

**Comment:**

We thank the authors for their time and contribution to WANT and we are pleased to share that after the reviewing process the paper has been accepted. Congratulations! We encourage the authors to consider reviewers' feedback for the improvement of the camera-ready version. We hope to see you in person at the workshop and brainstorm on efficient training research together!